# Study Rheological Behavior of Polymer Solution in Different-Medium-Injection-Tools

**DOI:** 10.3390/polym11020319

**Published:** 2019-02-13

**Authors:** Bin Huang, Xiaohui Li, Cheng Fu, Ying Wang, Haoran Cheng

**Affiliations:** 1The Key Laboratory of Enhanced Oil and Gas Recovery of Educational Ministry, Northeast Petroleum University, Daqing 163318, China; huangbin111@163.com (B.H.); lixiaohuibusiness@163.com (X.L.); 2Post-Doctoral Scientific Research Station, Daqing Oilfield, Daqing 163413, China; 3Aramco Asia, Beijing 100102, China; ying.wang.1@aramcoasia.com; 4Research Institute of Tsinghua University in Shenzhen, Shenzhen 518057, China; haoran.cheng@icore-group.com

**Keywords:** polymer flooding, different-medium-injection-tool, maximum injection velocity, apparent viscosity, polymer molecular weight

## Abstract

Previous studies showed the difficulty during polymer flooding and the low producing degree for the low permeability layer. To solve the problem, Daqing, the first oil company, puts forward the polymer-separate-layer-injection-technology which separates mass and pressure in a single pipe. This technology mainly increases the control range of injection pressure of fluid by using the annular de-pressure tool, and reasonably distributes the molecular weight of the polymer injected into the thin and poor layers through the shearing of the different-medium-injection-tools. This occurs, in order to take advantage of the shearing thinning property of polymer solution and avoid the energy loss caused by the turbulent flow of polymer solution due to excessive injection rate in different injection tools. Combining rheological property of polymer and local perturbation theory, a rheological model of polymer solution in different-medium-injection-tools is derived and the maximum injection velocity is determined. The ranges of polymer viscosity in different injection tools are mainly determined by the structures of the different injection tools. However, the value of polymer viscosity is mainly determined by the concentration of polymer solution. So, the relation between the molecular weight of polymer and the permeability of layers should be firstly determined, and then the structural parameter combination of the different-medium-injection-tool should be optimized. The results of the study are important for regulating polymer injection parameters in the oilfield which enhances the oil recovery with reduced the cost.

## 1. Introduction

Approximately 92% of reservoirs in China are continental sedimentation with clastic rock, accounting for more than 1/4 of the proven geological reserves. Their highly vertical heterogeneities lead to increasing difficulty in injection development. Polymer flooding has been used in Daqing oilfield to improve the contradiction between layers in the process of oilfield development [1,2,3]. The oil recovery can be increased 13% in comparison to that of water flooding [4]. In order to correlate the relationship between the molecular weight of different polymers and the permeability of different geological layers, the polymer-separate-layer-injection-technology, separates mass and pressure in a single pipe, was proposed in Daqing oilfield. The aim of the polymer-separate-layer-injection-technology is to inject high molecular weight polymer into a high permeability layer; for low permeability layer, the injection rate of polymer with low molecular weight should be increased. The technology can control both the polymer molecular weight and the injection volume to maximize the oil recovery and improve the polymer flooding efficiency. It mainly includes one partial-pressure-injection-tool and one different-medium-injection-tool. The main function of the different-medium-injection-tool is to distribute the polymer molecular weight in different thin oil layer. Since the molecular weight of a polymer can be characterized by its apparent viscosity [5], the loss of apparent viscosity is an important index to determine the properties of the different-injection-tool.

Santhosh K Veerabhadrappa [6] designed a laboratory experiment on the basis of Littmann and Khan’s [7,8] research on the degradation of polymer solution after mechanical shearing. Their studies on the rheological properties of polymer solution include the shear stress distributions, shear viscosity, viscous modulus and elastic modulus which were measured before and after the core flooding. At present, polymer flooding technology has been widely used in oilfield development and the study on rheological properties of polymers has also made significant progress. KS Sorbie [9] studied the mechanism of polymer flooding to improve the recovery ratio based on polymer structure, stability and solution behavior. It [10] improves the water injection profile by applying cross flow between the vertically heterogeneous layers and reduces the viscosity index to improve the recovery ratio. H Mahani [11] provided a practical method to infer the in-situ polymer rheology and the induced fracture dimensions from PFO tests performed during polymer injection. It combines the realistic polymer rheology which was proposed based on the analysis of pressure derivative curve of polymer rheological parameter method. This method also added the effects of fracture size on radial stable polymer as a point of innovation which makes it possible to obtain more precise polymer injection characteristics, formation properties, and information during injection. Santhosh K Veerabhadrappa’s experiments and the polymer were screened to be successful for field application. The experiment result shows that different molecular weights of polymer solution have different displacement efficiency for different permeability of oil layers. Therefore, the molecular weight of polymer should be selected to match the different permeability of oil layers [12,13].

Daqing is the first oil company which proposed the use of the polymer-separate-layer-injection-technology to improve the efficiency of single concentration of polymer solution matching different permeability of oil layers. Li et al. [14] compared the injection effects of three different injection separation processes with the traditional injection mode through the field experiments. The polymer-separate-layer-injection-technology was applied to nearly 800 wells on site. Compared with general injection wells, the utilization of oil layers was improved with the number of layers increased from 68.2% to 73.9% and the thickness increased from 73.9% to 80.9%. The results showed that the polymer-separate-layer-injection-technology can match the permeability of layers and the molecular weight of polymer and improve the efficiency of polymer injection. Compared with general injection mode, it has incomparable advantages. In order to overcome the limitations of the traditional displacement test of small core on the macro-scale, Li et al. [15] developed a three-layer non-homogeneous third-plate radial flow physical model. The comparison between the general injection and the polymer separate layer injection shows that polymer separate layer injection has obvious advantages over general injection. Wang et al. found that some Daqing wells with significant permeability differential between layers and no crossflow, injecting polymer solutions separately into different layers improved flow profiles, reservoir sweep efficiency, and injection rates [16]. The above research shows that the recovery efficiency of oil fields can be improved significantly by improving the rheology property of the polymer solution. Therefore, selecting the type of polymer and understanding how its influence by fluid rheology are probably among the most critical factors involved in designing a successful polymer flood. In oilfield application, the separate layer injection tools must be put into annulus to work. Because of the strength and stability of polymer molecular structure, only when the energy and frequency of shearing or turbulent pulsation reach a critical value can the structure of molecular mass be destroyed significantly. And turbulence is the randomness of the micelle fluid movement, turbulent micelle not only has transverse fluctuation and there is always relative to the fluid movement of the inverse kinematics, so fluid micelle trajectory is disordered, changed rapidly with time [17]. The transfer of momentum, heat and mass caused by this random motion has several orders of magnitude higher than that of laminar flow, and causing greatly increases friction resistance and energy loss [18,19].

So during polymer solution flowing through a different-medium-injection-tool, the shear thinning behavior of polymer solution should be used to reduce the flow resistance [20]. However, high injection speed should be avoided due to turbulent flow of polymer solution through the different-medium-injection-tool. So as to reduce energy loss and dissipation [21]. The polymer rheology in the flow process of different-medium-injection-tools can determine the largest polymer solution injection rate to improve oil recovery and reduce the development cost which has practical value in engineering [22].

In this paper, the velocity and apparent viscosity of polymer in circular tube in laminar state are derived from the law of motion and rheology of polymer. Based on the local perturbation theory, the model describing apparent viscosity of polymer solution under the condition of critical instability is derived. The concept of maximum injection velocity was proposed, and the mathematical model was extrapolated to calculate the apparent viscosity of turbulent polymer solution. The flow of polymer solution in different medium-injection-tool is considered as a flow process that expands in a pipe with variable cross-section, and then remains unchanged till the final contraction. A mathematical model of the rheological properties of the polymer solution in different-medium-injection-tool is established. The partially hydrolyzed polyacrylamide of 1600 × 10^4^ molecular weight used in Daqing oilfield were used for laboratory experiment at concentrations of 1000 mg/L, 1200 mg/L and 1500 mg/L. The power law curves of polymer solution were fitted. The mathematical models were used to study the rheological properties of polymer solution in different-medium-injection-tool and different injection rates.

## 2. Rheological Model of Polymer Solution in Circular Tube

The polymer solution is assumed to be an incompressible power law fluid.

The Rivlin-Ericksen tensor of power law fluid is shown below:
(1)$$A_{rr}=A_{\theta \theta }=A_{xx}=A_{r\theta }=0$$
(2)$$A_{rx}=-\frac{du}{dr}$$
where *A* is tensor; **u** is velocity vector, m/s; r is radius, m.

The constitutive equation for non-Newtonian power law fluid is shown below:
(3)$$\tau =k{\left[\frac{1}{2}\mathrm{tr}A_{1}^{2}\right]}^{\frac{n-1}{2}}A$$
where τ is deviator stress tensor.

Thus:
(4)$$\tau_{ij}=k{\left[\frac{1}{2}{\mathrm{trA}}_{im}A_{mj}\right]}^{\frac{n-1}{2}}A_{ij}$$


### 2.1. A Mathematical Model of Polymer Flow in A Circular Tube in Laminar Flow

When the polymer solution moves i in a circular tube in a laminar flow state:

The deviator stress tensor in the flow field satisfies the following relation:
(5)$$\tau_{rr}=\tau_{\theta \theta }=\tau_{xx}=\tau_{r\theta }=\tau_{\theta x}=0$$
(6)$$\tau_{rx}=k{\left|-\frac{du}{dr}\right|}^{n-1}\frac{du}{dr}=k{\left(\frac{du}{dr}\right)}^{n}$$


When the polymer solution is in laminar flow, the linear flow field is parallel to the axial direction, therefore:
(7)$$u_{r}=0\;u_{\theta }=0\;u_{x}=u\left(r\right)$$


According to the continuous equation of incompressible fluid in cylindrical coordinates:
(8)$$\frac{\partial u_{r}}{\partial r}+\frac{1}{r}\frac{\partial u_{\theta }}{\partial \theta }+\frac{\partial u_{x}}{\partial x}=0$$


Then:
(9)$$\frac{\partial u_{x}}{\partial x}=0$$


According to the Navier-Stokes equations for the incompressible fluid:
(10)$$\rho \frac{du}{dt}=\rho f-\nabla p+\mu \nabla^{2}u$$
where ρ is density, kg/m^3^; *p* is pressure, Pa.

Because of the polymer solution is an incompressible fluid, the N-S equation is simplified as:
(11)$$\begin{array}{l} \frac{\partial u_{r}}{\partial t}+u_{r}\frac{\partial u_{r}}{\partial r}-\frac{u_{\theta }}{r}+\frac{u_{\theta }}{r}\frac{\partial u_{r}}{\partial \theta }+u_{x}\frac{\partial u_{r}}{\partial x}=-\frac{1}{\rho }\frac{\partial p}{\partial r}+\nu \left(\nabla^{2}u_{r}-\frac{u_{r}}{r^{2}}-\frac{2}{r^{2}}\frac{\partial u_{\theta }}{\partial \theta }\right)\\ \frac{\partial u_{\theta }}{\partial t}+u_{r}\frac{\partial u_{\theta }}{\partial r}+\frac{u_{r}\cdot u_{\theta }}{r}+\frac{u_{\theta }}{r}\frac{\partial u_{\theta }}{\partial \theta }+u_{x}\frac{\partial u_{\theta }}{\partial x}=-\frac{1}{\rho \cdot r}\frac{\partial p}{\partial \theta }+\nu \left(\nabla^{2}u_{\theta }-\frac{u_{\theta }}{r^{2}}\frac{\partial^{2}u_{\theta }}{\partial r^{2}}+\frac{2}{r^{2}}\frac{\partial u_{r}}{\partial \theta }\right)\\ \frac{\partial u_{\theta }}{\partial t}+u_{r}\frac{\partial u_{x}}{\partial r}+\frac{u_{\theta }}{r}\frac{\partial u_{x}}{\partial \theta }+u_{x}\frac{\partial u_{x}}{\partial z}=-\frac{1}{\rho }\frac{\partial p}{\partial x}+\nu \nabla^{2}u_{x} \end{array}$$
where ν is kinematic viscosity, m^2^/s.

The fluid boundary conditions under the laminar flow are shown below:
(12)$$u_{r}=0\;u_{\theta }=0\;u_{x}=u\left(r\right)$$


Combined Equation (11) and Equation (12):
(13)$$\frac{\partial p}{\partial r}=0\;\frac{1}{r}\frac{\partial p}{\partial \theta }=0-\frac{\partial p}{\partial x}+\frac{\partial \tau_{rx}}{\partial r}=0$$


Then put Equation (9) into the Equation (13)
(14)$$\frac{\partial p}{\partial x}=\left(k{\left|\frac{\partial u}{\partial r}\right|}^{n-1}\frac{du}{dr}\right)$$


The flow is driven by pressure only, corresponding to:
(15)$$\frac{dp}{dx}<0$$
(16)$$\frac{du}{dr}<0$$


Integral from the pipe wall to the axis for Equation (15) (*r*_0_→*r*):
(17)$$p=C_{1}x+C_{2}$$


*C*_2_ = 0,
(18)$$k{\left(-\frac{du}{dr}\right)}^{n}=C_{1}r+C_{3}$$


According to the flow state, when the flow velocity changes from 0 to *u*(*r*), representing from the pipe wall to the center axis, thus, *C*_3_ = 0.

Integrate the Equation (18).
(19)$$k{\left(-\frac{du}{dr}\right)}^{n}=C_{1}r\Rightarrow u\left(r\right)=\frac{n}{n+1}{\left(\frac{\Delta p}{2kL}\right)}^{\frac{1}{n}}\left(r_{0}^{\frac{n+1}{n}}-r^{\frac{n+1}{n}}\right)$$
where L is length, m.

Because the fluid in laminar state means the equilibrium condition of forces:
(20)$$\tau \cdot 2\pi rL=\Delta p\cdot \pi r^{2}$$
(21)$$\tau =\frac{\Delta p\cdot r}{2L}=k\left(-\frac{du}{dr}\right)$$


It can be obtained from Equations (18)–(20):
(22)$$C_{1}=\frac{\Delta p}{2L}$$
(23)$$u\left(r\right)=\frac{n}{n+1}{\left(\frac{\Delta p}{2kL}\right)}^{\frac{1}{n}}\left(r_{0}^{\frac{n+1}{n}}-r^{\frac{n+1}{n}}\right)$$


From the above two equations, it can be obtained that the velocity of the power law fluid and the shear rate at the wall are related to the radius of the pipe and its position in the pipe. The conclusion is consistent with the winding theory for shear thinning of pseudoplastic [23] fluids. In the polymer solution, small molecules that have been encased in a particle or a macromolecular in a cavity due to solvation are forced out. Thus, with the increasing of stress, the effective diameter of a particles or hovering macromolecular decreases, so that the viscosity of fluid decreases. It enhances the fluid mobility and increases the velocity of polymer solution. The shear rate decreases gradually from the central axis of the pipe to the wall of the pipe. Therefore, the central velocity of the circular pipe is significantly higher than that of the wall. The shear rate is related to the radius and location of the pipe.

When r = 0, the polymer solution has the maximum velocity, which can be obtained:
(24)$$u_{max}=\frac{n}{n+1}{\left(\frac{\Delta p}{2kL}\right)}^{\frac{1}{n}}r_{0}^{\frac{n+1}{n}}$$


The average velocity at any vertical axial section of the pipeline can be expressed as:
(25)$$\bar{u}=\int_{0}^{r_{0}}\frac{u_{r}\cdot 2\pi rdr}{\pi r_{0}^{2}}$$


Or:
(26)$$\bar{u}=\frac{n+1}{3n+1}u_{\max}$$


In horizontal circular pipe, the frictional head loss of polymer solution can be expressed as:
(27)$$h_{f}=\frac{\nabla p}{\gamma }=\frac{64}{\frac{8^{1-n}\rho D^{n}{\bar{u}}^{2-n}}{k{\left(\frac{3n+1}{4n}\right)}^{n}}}\frac{L}{D}\frac{{\bar{u}}^{2}}{2g}\;D=2r_{0}$$
According to the definition of the frictional head loss, Reynolds number can be expressed as:
(28)Re=81−nρDnu¯2−nk(3n+14n)n
The shear rate at anywhere of the corresponding pipe section can be expressed as:
(29)$$\gamma_{r}=-\frac{du}{dr}={\left(\frac{\Delta p}{2LK}\right)}^{\frac{1}{n}}r^{\frac{1}{n}}=\frac{3n+1}{n}\frac{Q}{\pi {r_{0}}^{\frac{3n+1}{n}}}r^{\frac{1}{n}}$$


According to Equations (20) and (25), the average apparent viscosity of the section is:
(30)$$\bar{\mu }=\frac{\int_{0}^{r_{0}}\mu_{r}dr}{r_{0}}=K\frac{n}{2n-1}{\left[\frac{\left(3n+1\right)Q}{n\pi {r_{0}}^{\frac{3n+1}{n}}}\right]}^{n-1}{r_{0}}^{\frac{2n-1}{n}}$$


### 2.2. Model of Maximum Injection Rate of Polymer Solution

The stability parameter Z proposed by Ryan and Johnso [24] is used as a criterion for the flow state of non-newtonian fluids. The method shows that the initial point of turbulence can be determined under different n values of different flow behavior indexes. The critical value always remains the same, the Reynolds number is 2100, the stability of the parameter Z of the critical value is 808.

According to the above theoretical basis and the local disturbance theory [25], the model of apparent viscosity under critical instability is derived. The concepts of maximum injection velocity and maximum injection rate are proposed. The mathematical model is also applicable to the calculation of apparent viscosity of polymer solution holds at turbulent state.

The physical definition of Reynolds number is the ratio of the inertial force to the viscous force:
(31)Re=ρD⋅u(r)γ


It can be obtained from Equations (26) and (29):
(32)$$\frac{\rho D\cdot u\left(r\right)}{\gamma }=\frac{8^{1-n}\rho D^{n}{\bar{u}}^{2-n}}{k{\left(\frac{3n+1}{4n}\right)}^{n}}$$
When the *r* = 0.5D, *u*(*r*) = 0, Re = 0;When the *r* = 0; *u*(*r*) = 0, Re = 0.


According to Roll theorem, there is a maximum value of Re, and the radius of the unstable position is:
(33)$$r_{us}={\left(\frac{1}{n+2}\right)}^{\frac{n}{n+1}}r_{0}$$


The velocity of the unstable position is shown below:
(34)$$u_{us}\left(r\right)=k\frac{n}{2n-1}\left\{\frac{\left(3n+1\right)Q}{n\pi \left[{\left(\frac{1}{n+2}\right)}^{\frac{2n+1}{n+1}}\right]}\right\}\frac{1}{r_{0}^{\frac{\left(2n+1\right)\left(n-1\right)}{n}}}$$


The apparent viscosity where the unstable position is located is shown below:
(35)$$\bar{\mu }=K\frac{n}{2n-1}{\left[\left(3n+1\right){\left(n+2\right)}^{\frac{2n+1}{n+1}}\right]}^{n-1}{\left(\frac{\bar{u}}{r_{0}}\right)}^{n-1}$$


The average velocity of polymer solution in turbulent flow can be calculated by the following equation:
(36)$$\bar{u}=\frac{Q}{A}$$


Therefore, the average apparent viscosity of the section at the critical position is shown below:
(37)$$\bar{\mu }=C{\left(\frac{\bar{u}}{r_{0}}\right)}^{n-1}$$


The constant C is calculated using the Equation (27):
(38)$$C=K\frac{n}{2n-1}{\left[\left(3n+1\right){\left(n+2\right)}^{\frac{2n+1}{n+1}}\right]}^{n-1}$$


The maximum injection velocity of laminar state at critical instability is shown below:
(39)$$\bar{u}={\left[\frac{2100K{\left(\frac{3n+1}{4n}\right)}^{n}}{8^{1-n}\rho D^{n}}\right]}^{\frac{1}{2-n}}$$


After the unstable flow of polymer solution, the laminar flow changed to turbulent flow. From the equation of instable apparent viscosity, the model is in accordance with the basic physical definition of apparent viscosity. The corresponding average velocity is the unstable velocity at which the laminar flow transforms to turbulent flow, called the maximum injection velocity of laminar flow which is very important to control the flux through different-medium-injection-tools. When the velocity exceeds the critical maximum flow rate, the polymer solution is in turbulent state. The above equation does not calculate the energy loss of turbulence and the average of viscosity loss.

## 3. Mathematical Model of the Polymer Flow in the Different-Medium-Injection-Tool

Polymer flooding improves the water injection profile in heterogeneous reservoirs by reducing the efficiency of fluid viscosity fingering between layers and the permeability contrast in order to enable the relative permeability of water phase greater than that of the oil phase [26]. The study on rheological properties of polymer solution injected into oil layers is an important step to increase displacement efficiency [27]. In the following part, based on the rheology of polymer solution under circular pipe condition, the rheology of polymer solution in different-medium-injection-tools is studied. The effects of different polymer solutions on the maximum injection velocity are analyzed. The injection rate and injection velocity of polymer solutions are suggested through the method of using different-medium-injection-tool in oil field. Different-medium-injection-tool model is shown in Figure 1.

The flow process of polymer solution in different-medium-injection-tool is the same as the process of contractive-tube flow-expansion. It is assumed that the fluid flow in the different-medium-injection-tool is unitary and isothermal and incompressible ignoring gravity and transition effect in the flow process. Shear thinning of polymer solution should be used to reduce the flow resistance. High injection speed should be avoided for causing the polymer solution to flow through the different-medium-injection-tool in a turbulent state and the energy loss and dissipation when injecting polymer solution into different-medium-injection-tools. Therefore, the polymer solution in the different-medium-injection-tool is laminar flow. The diagram shows the cross-section of a different-medium-injection-tool model in Figure 2.

Where *L*_1_ is the length of the contraction section of the different-medium-injection-tool; *L*_2_ is the length of circular tube of the different-medium-injection-tool; *L*_3_ is the length of diffused section of the different-medium-injection-tool; R is the radius of inlet of contraction section; *r*_1_ is the distance between the axis of the contraction section and the surface of the tool; *r*_2_ is the distance between the axis of the circular tube and the surface of the tool; *r*_3_ is the distance between the axis of the diffused section and the surface of the tool; d is the diameter of the circular tube; D is the maximum diameter of the diffused section.

The shear rate and shear stress of the fluid unit at the different-medium-injection-tool can be approximately expressed by the average shear rate and shear stress in the circular tube.

At the *X*-axis direction, when 0 < *x* < *L*_1_, the distance between the axis of the contraction section and the surface of the tool, the velocity distribution and the apparent viscosity distribution in the contraction section of the different-medium-injection-tool can be expressed as follows:
(40)$$r_{1}=\left(\frac{{\left(R-\left(\frac{d}{2}\right)\right)}^{2}+L_{1}^{2}}{2\left(R-\left(\frac{d}{2}\right)\right)}+\frac{d}{2}\right)-\sqrt{{\left(\frac{{\left(R-\left(\frac{d}{2}\right)\right)}^{2}+L_{1}^{2}}{2\left(R-\left(\frac{d}{2}\right)\right)}\right)}^{2}-{\left(x-L_{1}\right)}^{2}}$$
(41)$$u_{1}=\frac{n}{3n+1}{\left(\frac{\Delta p}{2kL}\right)}^{\frac{1}{n}}{\left(\left(\frac{{\left(R-\left(\frac{d}{2}\right)\right)}^{2}+L_{1}^{2}}{2\left(R-\left(\frac{d}{2}\right)\right)}+\frac{d}{2}\right)-\sqrt{{\left(\frac{{\left(R-\left(\frac{d}{2}\right)\right)}^{2}+L_{1}^{2}}{2\left(R-\left(\frac{d}{2}\right)\right)}\right)}^{2}-{\left(x-L_{1}\right)}^{2}}\right)}^{\frac{n+1}{n}}$$
(42)$${\bar{\mu }}_{1}=K\frac{n}{2n-1}{\left[\frac{\left(3n+1\right)Q}{n\pi {\left(\left(\frac{{\left(R-\left(\frac{d}{2}\right)\right)}^{2}+L_{1}^{2}}{2\left(R-\left(\frac{d}{2}\right)\right)}+\frac{d}{2}\right)-\sqrt{{\left(\frac{{\left(R-\left(\frac{d}{2}\right)\right)}^{2}+L_{1}^{2}}{2\left(R-\left(\frac{d}{2}\right)\right)}\right)}^{2}-{\left(x-L_{1}\right)}^{2}}\right)}^{\frac{3n+1}{n}}}\right]}^{n-1}{\left(\left(\frac{{\left(R-\left(\frac{d}{2}\right)\right)}^{2}+L_{1}^{2}}{2\left(R-\left(\frac{d}{2}\right)\right)}+\frac{d}{2}\right)-\sqrt{{\left(\frac{{\left(R-\left(\frac{d}{2}\right)\right)}^{2}+L_{1}^{2}}{2\left(R-\left(\frac{d}{2}\right)\right)}\right)}^{2}-{\left(x-L_{1}\right)}^{2}}\right)}^{\frac{2n-1}{n}}$$


At the *X*-axis direction, when *L*_1_< *x* < *L*_2_, the distance between the axis of the circular tube and the surface of the tool, the velocity distribution and the apparent viscosity distribution in the contraction section of the different-medium-injection-tool can be expressed as follows:
(43)$$r_{2}=\frac{d}{2}$$
(44)$$u_{2}=\frac{n}{3n+1}{\left(\frac{\Delta p}{2kL}\right)}^{\frac{1}{n}}{\left(\frac{d}{2}\right)}^{\frac{n+1}{n}}$$
(45)$${\bar{\mu }}_{2}=K\frac{n}{2n-1}{\left[\frac{\left(3n+1\right)Q}{n\pi {\left(\frac{d}{2}\right)}^{\frac{3n+1}{n}}}\right]}^{n-1}{\left(\frac{d}{2}\right)}^{\frac{2n-1}{n}}$$


At the *X*-axis direction, when *L*_2_< *x* < *L*_3_, the distance between the axis of the contraction section and the surface of the tool, the velocity distribution and the apparent viscosity distribution in the diffused section of the different-medium-injection-tool can be expressed as follows:
(46)$$r_{3}=\frac{d}{2}+\frac{D-d}{2L_{3}}x$$
(47)$$u_{3}=\frac{n}{3n+1}{\left(\frac{\Delta p}{2kL}\right)}^{\frac{1}{n}}{\left(\frac{d}{2}+\frac{D-d}{2L_{3}}x\right)}^{\frac{n+1}{n}}$$
(48)$${\bar{\mu }}_{3}=K\frac{n}{2n-1}{\left[\frac{\left(3n+1\right)Q}{n\pi {\left(\frac{d}{2}+\frac{D-d}{2L_{3}}x\right)}^{\frac{3n+1}{n}}}\right]}^{n-1}{\left(\frac{d}{2}+\frac{D-d}{2L_{3}}x\right)}^{\frac{2n-1}{n}}$$


## 4. The Flow of Polymer Solution in the Different-Medium-Injection-Tool-Daqing Oilfield

Experimental supplies and methods.

The water used for polymer configuration is the water after treatment in No. 1 Oil Production Plant in Daqing Oilfield, and the ion compositions of water were shown in Table 1.

Experimental instruments: Thermostatic water bath, S 501-2; Huaguang instrument factory in Liaoyang, Liaoyang, China; Agitator, EURO-ST D S25 from the IKA Company in Germany, Guangzhou, China; Rheometer, AR2000ex, NewCastle, PA, USA.

Experimental materials partially hydrolyzed polyacrylamide with 1600 × 10^4^ molecular weight (HPAM) from Petrochemical Plant of Daqing Petroleum, Daqing, China.

Experimental method:

(1) The polymer mother liquor and the target liquid were prepared on the basis of “Recommended practices for evaluation of polymers used in enhanced oil recovery” (SY/T 6576-2016), and the concentration of mother liquor is 5000 mL/L.

(2) The polymer mother liquor was mixed and diluted with water, and the concentrations of polymer solution were 1000 mg/L, 1500 mg/L and 2000 mg/L, respectively, and the polymer solution were settled for 6 h to ensure the system uniform.

(3) The shear test for the polymer solution was conducted at a water bath temperature of 45 °C to measure the rheological curve, and the shear rate was from 1 to 1000 s^−1^.

Experimental result.

The experimental results are plotted in Figure 3.

Figure 3 shows the relationship between the shear rate and apparent viscosity of polymer solutions at different concentrations. It can be seen the higher of the concentration at the same shear rate, the greater the viscosity of polymer solutions. The apparent viscosity of polymer solutions decreases with the increase of shear rate. The power law equation is fitted the experimental results as shown in Table 2. The consistency coefficient K increases with the increase of polymer concentration. The liquidity index n is opposite to that of K. With the increase of polymer solution concentration, n decreases and non-newtonian property of the polymer solution increases. This experimental result is consistent with the measurement results of Necmettin Mungan and Kong et al. [28,29], which proves the polymer solution is power law fluid.

The power law equation of HPAM solution was applied to the mathematical model to study the rheological properties of the polymer solution in the different-medium-injection-tool. Table 3 shows the structural parameters of the different-medium-injection-tool commonly used in Daqing oilfield.

The maximum injection velocity should be determined first after the structural parameters of the different-medium-injection-tool and the power law curves of the polymer solution were determined. According to the above conditions, the maximum injection rate of HPAM solution of 1000 mg/L, 1500 mg/L and 2000 mg/L was calculated respectively.

According to Equation (36), the corresponding maximum injection velocity is 24.36 m/s, 36.44 m/s and 85.09 m/s respectively. The maximum injection rate is 26.44 m^3^/d, 39.54 m^3^/d, and 92.33 m^3^/d, respectively. The maximum injection velocity and the maximum injection rate of polymer solution increased with the concentration of polymer solution which is caused by the non-newtonian property of polymer solution enhanced, the molecular coil of polymer solution relatively more dense and tightly wound, and the structural strength and stability of the molecular coil increased with the increase of concentration. When the polymer solutions with different concentrations are subjected to the same energy of destruction, the polymer solutions with higher concentrations are more stable. The maximum injection velocity and injection rate are larger.

When polymer solutions of three different concentrations flow through the different-medium-injection-tool for the same structural parameter at the injection rate of 20 m^3^/d, polymer solutions flow in a laminar state through the different-medium-injection-tool. Figure 4 shows the relationship between the velocity of polymer solution with different concentration and location in the different-medium-injection-tool. It can be seen that the velocity of polymer solution at different locations in the different-medium-injection-tool is: first increasing, then not changing, and last declining. This is caused by the change of the radius of each part of the different-medium-injection-tool. The velocity in the different-medium-injection-tool is increasing with the concentration of HPAM solution. The reason is that as the concentration of polymer solution increases, the non-Newtonian behavior gradually increases, and flow resistance increases, resulting in a significant decrease in flow velocity. However, the radius of the outlet of the different-medium-injection-tool is less than the inlet radius, so the outlet velocity is higher than the inlet velocity. When the injection rate is satisfied, the outlet velocity is higher and the pressure drop is smaller for the HPAM solution of 1000 mg/L, which has a positive effect on increasing the effect of the polymer separate layer injection.

Figure 5 shows the relationship between the apparent viscosity of polymer solution with different concentration and location in the different-medium-injection-tool. The apparent viscosity of the polymer solution is completely opposite to the velocity changes. During the flow of polymer solution into the tool, the apparent viscosity of the solution decreases sharply, and the instantaneous apparent viscosity of polymer solution into the tool decreases the most. It can be seen that the apparent viscosity of polymer solution at different location of the different-medium-injection-tool is: first declining, then not changing, and last rising. This is caused by the change of the radius of each part of the different-medium-injection-tool which leads to the change of the apparent viscosity of the polymer solution. The apparent viscosity in the different-medium-injection-tool is increasing with the concentration of HPAM solution. In the contraction section, the apparent viscosity of polymer solution decreases gradually due to the circular shape of the boundary. The apparent viscosity of polymer solution remained stable because the shear stress remained constant in the circular section. When the polymer solution enters the diffusion section, the apparent viscosity recovers with the increase of the distance of the diffusion section. The range of change of the viscosity of the polymer solution in HPAM solution of 1000 mg/L, 1500 mg/L and 2000 mg/L is 8.8%, 9.5% and 13.2% respectively. By comparing the apparent viscosity change range of polymer solutions at three concentrations, it can be seen that with the increasing of mass concentration, the apparent viscosity of polymer solutions in the different-medium-injection-tool gradually increases. The higher concentration of polymer solution, the change of the apparent viscosity is more obvious of the different-medium-injection-tool with the same structural parameters. It can be seen from the above comparison that the variation range of viscosity of polymer solution in the different-medium-injection-tool is mainly determined by the structural parameters of the different-medium-injection-tool. However, the value of the viscosity of polymer solution is mainly determined by the concentration of polymer solution. In the application of polymer flooding, the relation between the molecular weight of polymer solution and the permeability of layers should be firstly determined for the viscosity requirement of the polymer solution, and then the structural parameter combination of the different-medium-injection-tool should be optimized to control the viscosity change range of polymer solution. 

## 5. Conclusions

Based on the rheology of the polymer and the local perturbation theory, the velocity distribution models and the apparent viscosity distribution models in the different-medium-injection-tool are derived. Combined with the experimental application of the model, the flow of polymer solution in the different-medium-injection-tool was analyzed. Based on the results, the following conclusions were drawn.

(1) During the polymer solution flowing through the different-medium-injection-tool, the maximum injection velocity and the maximum injection rate of laminar flow exist. The maximum injection velocity of the HPAM solutions with different concentrations of 1000 mg/L, 1500 mg/L, and 2000 mg/L is 24.36 m/s, 36.44 m/s and 85.09 m/s respectively. The maximum injection rate is 26.44 m^3^/d, 39.54 m^3^/d, and 92.33 m^3^/d, respectively.

(2) The distribution of the velocity of polymer solution at different locations of the different-medium-injection-tool is: increasing first, then not changing, and declining at last. When the injection rate is satisfied, the outlet velocity is higher and the pressure drop is smaller for the HPAM solution of 1000 mg/L, which exercises a positive impact on increasing the effect of stratified injection.

(3) The apparent viscosity of polymer solution at different locations in the different-medium-injection-tool is: declining first, then not changing, and last increasing. The instantaneous apparent viscosity of polymer solution into the tool decreases the most. 

(4) The range of variation for the viscosity of polymer solution in the different-medium-injection-tool is mainly determined by the structural parameters of the different-medium-injection-tool. However, the value of the viscosity of polymer solution is mainly determined by the concentration of polymer solution.

In this paper, the rheological model of polymer solution is derived based on the study of polymer properties. Without field experiment, the model can be verified and improved on the basis of field test in the future if conditions permit.

## Figures and Tables

**Figure 1 polymers-11-00319-f001:**
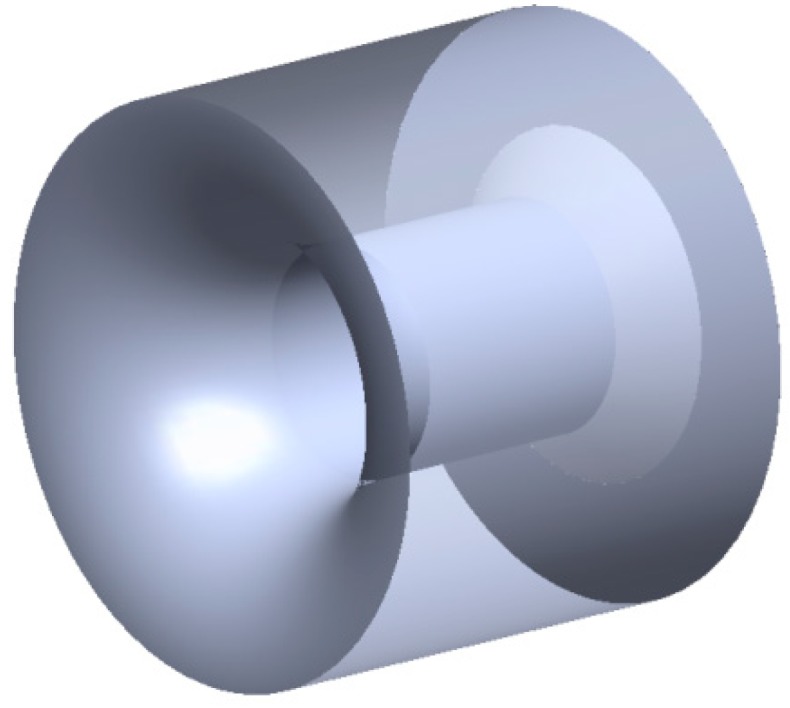
The model showing different-medium-injection-tool.

**Figure 2 polymers-11-00319-f002:**
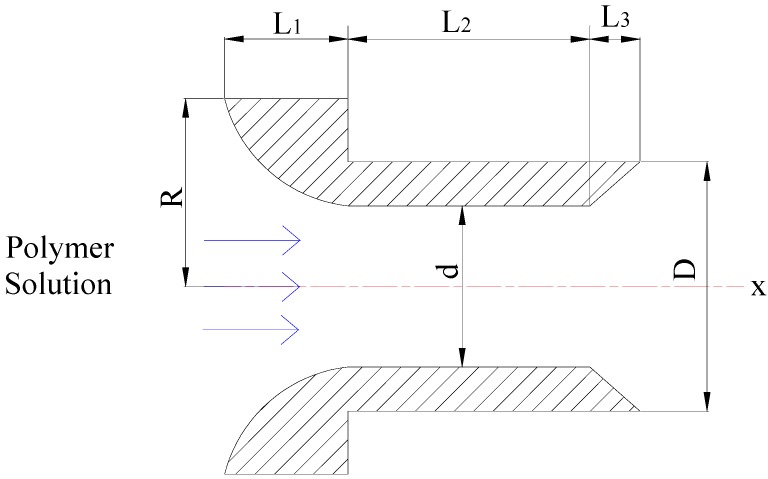
The cutaway view of a different-medium-injection-tool.

**Figure 3 polymers-11-00319-f003:**
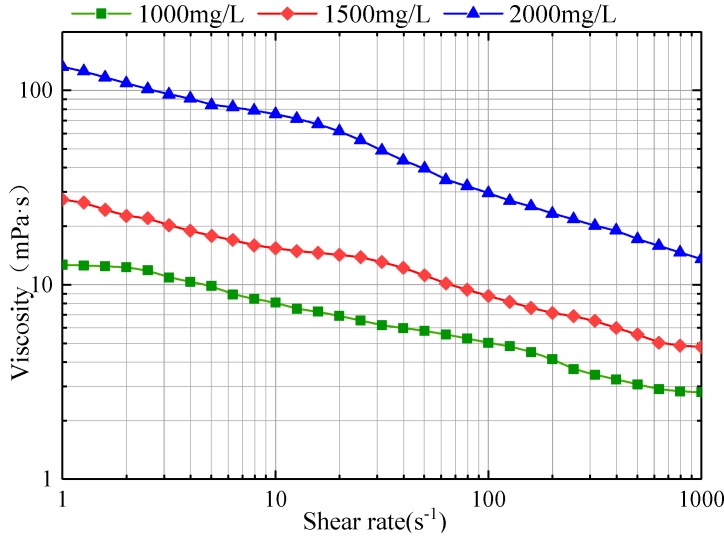
Relationship between shear rate and apparent viscosity of polymer solution at different concentrations.

**Figure 4 polymers-11-00319-f004:**
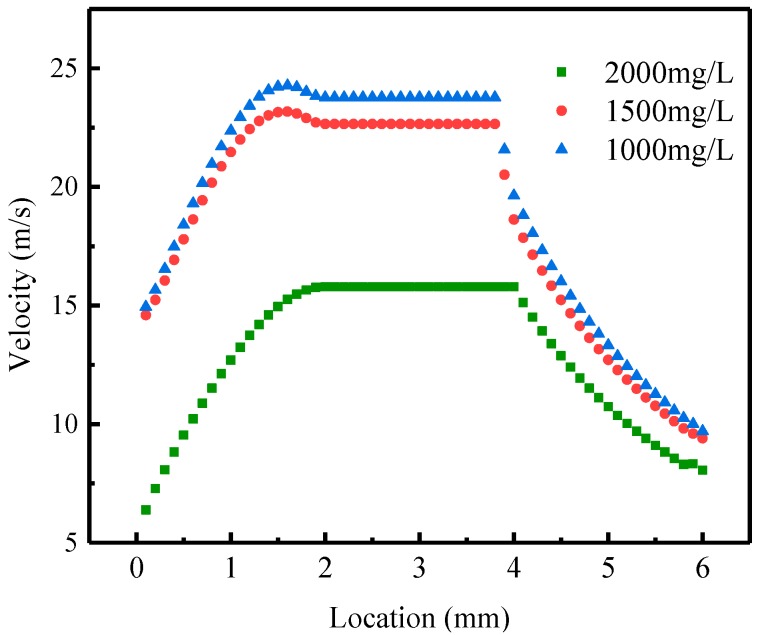
The relationship between the velocity of polymer solution of different concentration and location in the different-medium-injection-tool.

**Figure 5 polymers-11-00319-f005:**
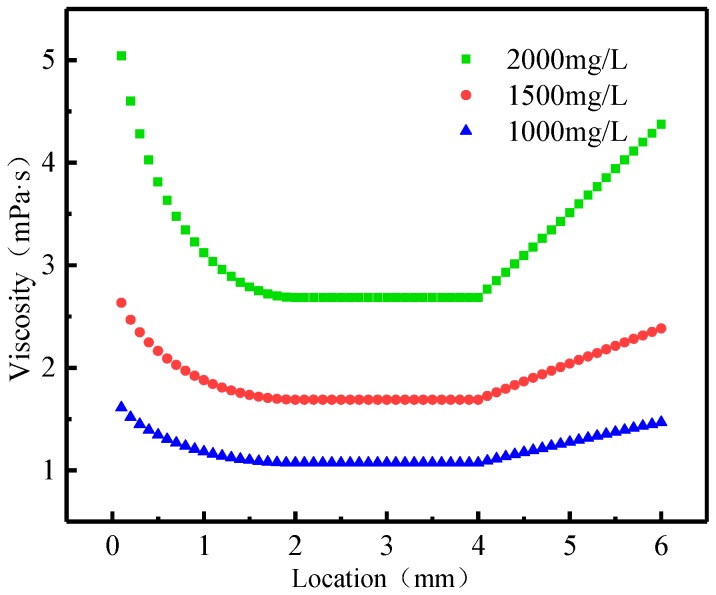
The relationship between the apparent viscosity of polymer solution with different concentration and location in the different-medium-injection-tool.

**Table 1 polymers-11-00319-t001:** The ion compositions of water.

Cl^−^	Na^+^, K^+^	Ca^2+^	Mg^2+^	SO_4_^2−^	HCO_3_^−^	CO_3_^2−^
53.1	50.6	28.06	7.29	9.61	30.01	126.28

**Table 2 polymers-11-00319-t002:** Power law equation of polymer at different concentrations.

Concentration	K	n	Power Law Equation
1000 mg/L	14.01	0.765	$\mu =14.01\gamma^{-0.235}$
1500 mg/L	28.14	0.746	$\mu =28.14\gamma^{-0.254}$
2000 mg/L	147.94	0.658	$\mu =147.94\gamma^{-0.342}$

**Table 3 polymers-11-00319-t003:** Structure parameters and injection parameters of the different-medium-injection-tool.

Contraction Radius R (mm)	Contraction Length *L*_1_ (mm)	Cylinder Diameter d (mm)	Cylinder Length *L*_2_ (mm)	Diffusion Length *L*_3_ (mm)	Maximum Diameter of Diffusion Section D (mm)
4	2	4	2	1	6

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
