# Peer review of "Study Rheological Behavior of Polymer Solution in Different-Medium-Injection-Tools"

_polymers, 2019, doi:10.3390/polym11020319_

Round 1

Reviewer 1 Report

The paper is interesting and of good quality, however this paper is not suitable for a this journal. The content of the manuscript is good for a journal of Chemical Engineering. The modelling is interesting but is not necessaryly related to polymers. In the application studied the polymers paly the role of thickeners, and the same goud be valid if other components were used, e.g. micro- or nanosuspensions.

I recommend to submit the manuscript to a journal more focused on oil engineering.

Author Response

Dear reviewer:

First of all, thank you very much for your comments on this article. We tried our best to improve the manuscript and made some changes in the manuscript. These changes are marked in the revised paper and will not affect the content and framework of the paper. We have submitted the word files of our revised manuscript and the replies to the comments of you. We appreciate for your warm work earnestly, and hope that the correction will meet with approval. Once again, thank you very much for your comments and suggestions.

Best regards,

Reviewer 2 Report

The manuscript entitled "Study Rheological Behavior of Polymer Solution in Different-medium-injection-tools" reports about an interesting study regarding the prevision of the polymer injection parameters in oilfield. in my opinion, the manuscript deserves to be published in Polymers.

I suggest to the authors to modify the Abstract, in order to enhance its readability, and to enriche the reference list with more recent papers.

Author Response

Dear reviewer:

First of all, thank you very much for your comments on this article. We tried our best to improve the manuscript according to your comments and made some changes in the manuscript. These changes are marked in the revised paper and will not affect the content and framework of the paper. We have submitted the word files of our revised manuscript and the replies to the comments of you. We appreciate for your warm work earnestly, and hope that the correction will meet with approval. Once again, thank you very much for your comments and suggestions.

Best regards,

Cheng Fu

Reviewer 3 Report

The manuscript entitled: "Study Rheological Behavior of Polymer Solution in Different-medium-injection-tools" is a work dealing with the problem of polymer solution flow during extrusion with a variety of injection tools.

The work has a solid theoretical background which is general and tries to cover a variety of aspects. I have nothing against the scientific results besides this work; I am just skeptical over the actual application of all the mentioned results.

Can the authors explain, why this method will be helpful to an experimentalist who will try to extrude of a specific polymer (e.g., a high molecular weight polyolefin with the problem of solubility at low temperatures)? 

Is it possible to specify what will happen to the model for different polymer types?

Author Response

Dear reviewer:

 First of all, thank you very much for your comments on this article. We tried our best to improve the manuscript and made some changes in the manuscript. These changes are marked in the revised paper and will not affect the content and framework of the paper. We have submitted the word files of our revised manuscript and the replies to the comments of you. We appreciate for your warm work earnestly, and hope that the correction will meet with approval. Once again, thank you very much for your comments and suggestions.

 Best regards,

 Cheng Fu

Round 2

Reviewer 1 Report

The reviewed version can be accepted for publication.